# Structured transverse orbital angular momentum probed by a levitated optomechanical sensor

Yanhui Hu [1,2], Jack J. Kingsley-Smith [1,2], Maryam Nikkhou[1,2], James A. Sabin[1,2], Francisco J. Rodríguez-Fortuño [1,2], Xiaohao Xu [3] ✉ & James Millen [1,2] ✉

The momentum carried by structured light fields exhibits a rich array of surprising features. In this work, we generate *transverse* orbital angular momentum (TOAM) in the interference field of two parallel and counter-propagating linearly-polarised focused beams, synthesising an array of identical handedness vortices carrying intrinsic TOAM. We explore this structured light field using an optomechanical sensor, consisting of an optically levitated silicon nanorod, whose rotation is a probe of the optical angular momentum, which generates an exceptionally large torque. This simple creation and direct observation of TOAM will have applications in studies of fundamental physics, the optical manipulation of matter and quantum optomechanics.

Our ability to control the momentum of light, and its transfer to matter, continues to deliver profound technological innovation. Optical confinement and cooling of atoms, earning a Nobel Prize for Chu, Cohen-Tannoudki & Phillips in 1997[1], is the core of many quantum technologies, and optical tweezers, which earned Ashkin a share of the 2018 Nobel Prize in Physics[2], have wide-ranging applications in healthcare, biophysics, and nanotechnology. Optical fields can carry a rich variety of forms of momenta, including *optical angular momentum* which can drive rotations via light-matter interactions[3].

Optical angular momentum itself manifests in many forms. The first distinction is drawn between spin angular momentum (SAM) and orbital angular momentum (OAM). The former originates from the rotation of the electromagnetic field vector, and the latter is the direct analogue of the classical angular momentum defined by $\mathbf{L} = \mathbf{r} \times \mathbf{P}$, where $\mathbf{r}$ is the displacement from the coordinate origin and $\mathbf{P}$ is the optical linear momentum density[4]. OAM seems, by definition, to be dependent on the choice of the coordinate system, and so can be called *extrinsic*. However, *intrinsic* (coordinate-independent) OAM can be obtained when the integral of the OAM density over space yields a non-zero value regardless of the coordinates chosen. This is the case in optical vortex beams, which exhibit OAM through their wavefronts spiralling around a phase singularity[5,6]. For paraxial light beams, the intrinsic OAM is *longitudinal* since $\mathbf{L}$ is parallel to the beam's

propagation direction. It is also possible to engineer *transverse* SAM, via evanescent waves[7,8], focused beams[9] and multiple wave interference patterns[10]. Transverse orbital angular momentum (TOAM) is rarely observed, unless we are considering polychromatic fields[11–13] or extrinsic momentum when a beam propagates away from the coordinate origin[14,15]. *Intrinsic* TOAM in monochromatic fields, appearing as intricate transverse phase singularity vortex lines, has been theoretically described in the near-field of the diffraction by sharp obstacles or slits[16,17] and in superimposed co-propagating beams with different beam-widths and/or amplitudes[18–20].

In this work, we present the first realisation of intrinsic TOAM. We create an optical angular momentum structure using two monochromatic offset counter-propagating beams, which carries both transverse SAM and intrinsic TOAM. The light field contains a robust array of synthesised transverse optical vortices. We verify and probe the optical angular momentum structure using a levitated nanoparticle optomechanical sensor[21], and demonstrate the tuneable nature of the induced torque.

The rotation of levitated nanoparticles via *longitudinal* SAM has recently been the focus of several studies[22–27]. In this work, we produce a torque five orders of magnitude larger than previously demonstrated, driving MHz rotation at 10 mbar of background gas pressure. The rotation of levitated particles has applications in torque

[1]Department of Physics, King's College London, Strand, London WC2R 2LS, United Kingdom. [2]London Centre for Nanotechnology, Department of Physics, King's College London, Strand, London WC2R 2LS, United Kingdom. [3]State Key Laboratory of Transient Optics and Photonics, Xi'an Institute of Optics and Precision Mechanics, Chinese Academy of Sciences, Xi'an 710119, China. ✉e-mail: xuxhao_dakuren@163.com; james.millen@kcl.ac.uk

sensing[23,26], studies of vacuum friction[28,29] and the exploration of macroscopic quantum physics[30,31]. The orbit of nanospheres via longitudinal OAM has been studied[32], and the presence of transverse SAM in focused circularly polarised light incident upon a microparticle has been inferred[33].

Our work presents a straightforward and robust method for generating intrinsic TOAM, the use of levitated nanoparticles as sensitive probes of structured light fields, and the first manipulation of particle motion using TOAM. The ability to fully control the alignment and rotation of nanoparticles levitated in vacuum is of great importance for cavity optomechanics[34], alignment of targets in high-energy beam experiments and quantum control at the nanoscale[31].

## Results

### Origin of the TOAM

We consider an illumination geometry consisting of two counter-propagating Gaussian beams with the same frequency $\omega$ and both linearly polarised in the $y$ direction (see Fig. 1). The waist plane of each beam is located at $z = 0$, and the positions of their axes are $(0, \pm\delta, 0)$. Assuming the time-dependence factor $e^{-i\omega t}$, one may analytically express the transverse electric and magnetic fields as,

$$
\begin{aligned}
E_y &= u(x, y + \delta, z)e^{ikz} + u(x, y - \delta, -z)e^{-ikz}, \\
B_x &= -\frac{1}{c}u(x, y + \delta, z)e^{ikz} + \frac{1}{c}u(x, y - \delta, -z)e^{-ikz},
\end{aligned}
\tag{1}
$$

where

$$
u(x, y, z) = E_0 \frac{w_0}{w(z)} e^{-i\varphi(z)} e^{\frac{ik(x^2 + y^2)}{2q(z)}},
\tag{2}
$$

is the Gaussian beam solution to the scalar Helmholtz equation in the slow-varying approximation[35]. Here, $E_0$ is a constant field amplitude, $k = 2\pi/\lambda$ is the wavenumber, $w_0 = w(z = 0)$ is the beam waist radius, $q(z) = z - iz_0$ and $\varphi(z) = \tan^{-1}(z/z_0)$ the Gouy phase, with $z_0 = (\pi w_0^2)/\lambda$ being the Rayleigh range.

Optical vortices stem from phase dislocations[36]. It can be proven (see Methods) that the phase dislocations of $E_y$ and $B_x$ in Eq. (1) occur when the beam offset is non-zero ($2\delta \neq 0$), along a series of singularity lines which lie on the $y = 0$ plane and take the values of $(x, z)$ that satisfy the following equation,

$$
\frac{k(x^2 + \delta^2)z}{z^2 + z_0^2} - \varphi(z) + kz = \frac{n\pi}{2},
\tag{3}
$$

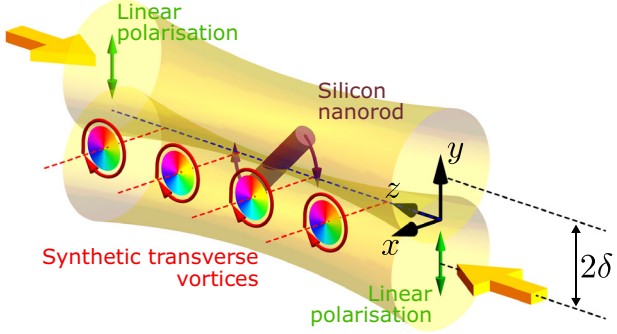

**Fig. 1 | Generation of structured transverse orbital angular momentum.** When two focused counter-propagating linearly polarised Gaussian beams are separated along the polarisation axis, an array of optical vortices is generated, carrying angular momentum which is transverse to the propagation direction. A silicon nanorod is suspended within the structure, generating a torque and driving rotation in the $y$–$z$ plane.

where $n$ takes odd and even integer values for electric and magnetic field singularities, respectively. This constitutes an array of alternating electric and magnetic field singularities.

As an illustration, Fig. 2 shows the calculated field characteristics for a beam offset $2\delta = 1.0$ μm; the wavelength and waist radius are set to $\lambda = 1550$ nm and $w_0 = 1.0$ μm, corresponding to beams focused with numerical aperture NA = 0.6. Both the electric and magnetic fields exhibit an intensity profile typical of standing waves: the positions of the electric field nodes (Fig. 2a) coincide with those of the magnetic field antinodes (Fig. 2b). The lines passing through the planes represent the location of the phase dislocations, namely, the solutions to Eq. (3) for $-6 \leq n \leq 6$. They stretch transversely along the $x$ direction, and pass through the nodes of the electric and magnetic fields for odd and even $n$, respectively.

From the phase profile (Fig. 2c), we can clearly identify the dislocation points for the electric and magnetic fields. For each point, the strength (a.k.a., topological charge) is $-1$, because the phase increases by $2\pi$ in a negative circuit with respect to the $+x$ direction. Therefore, the field carries a net OAM in the $-x$ direction. The black and white arrows show the electric $\mathbf{P}^e$ and magnetic $\mathbf{P}^m$ parts of the orbital momentum density[4]:

$$
\mathbf{P} = \mathbf{P}^e + \mathbf{P}^m = \frac{1}{4\omega}\Im\left\{\varepsilon_0 \mathbf{E}^* \cdot (\nabla)\mathbf{E} + \mu_0^{-1}\mathbf{B}^* \cdot (\nabla)\mathbf{B}\right\},
\tag{4}
$$

which, together with the position vector $\mathbf{r}$, defines the density of OAM: $\mathbf{L} = \mathbf{r} \times \mathbf{P}$. Around the dislocation points, $\mathbf{P}^e$ and $\mathbf{P}^m$ circulate in the same sense, such that all the phase vortices are of the same handedness. The existence of TOAM in this illumination is simple to understand if we imagine an extended particle, such as our rod, placed at the origin (Fig. 1): roughly speaking, opposite ends of the rod will be pushed (via an optical pressure force caused by each beam's orbital momentum density $\mathbf{P}$) in opposite directions, generating a torque on the particle. This torque is therefore enabled by the transverse nature of the OAM density (see Supplementary Information Section A & B, and Supplementary Movie 1 for more details).

The *integral* OAM carried by the field, $\langle\mathbf{L}\rangle$, is intrinsic since it is independent of the choice of origin; for a translational transformation of the coordinates $\mathbf{r} \to \mathbf{r} + \mathbf{r}_0$, there is no change since $\langle\mathbf{L}\rangle \to \langle\mathbf{L}\rangle + \mathbf{r}_0 \times \langle\mathbf{P}\rangle = \langle\mathbf{L}\rangle$, where we invoke that the integral momentum $\langle\mathbf{P}\rangle = 0$ due to the counter-propagating configuration of the two beams. Each of the Gaussian beams, considered separately, would possess a transverse extrinsic TOAM due to their propagation axis not crossing the origin. However, when taken together, their common centroid lies exactly in the origin, and no extrinsic TOAM exists. The extrinsic TOAM that would be carried by the beams if considered separately is expressed as intrinsic TOAM when the beams are taken together, and readily manifests itself in the appearance of the robust array of transverse phase vortices - for this reason we describe the vortices carrying intrinsic TOAM as synthetic.

So far we have not considered the longitudinal fields which, according to Maxwell equations, can be expressed in the post-paraxial approximation as,

$$
E_z \approx -\frac{ic^2}{\omega}\frac{\partial B_x}{\partial y}, \quad B_z \approx -\frac{i}{\omega}\frac{\partial E_y}{\partial x}.
\tag{5}
$$

These would be negligible in the case of weak focusing, however in moderate or strong focusing, they provide further interesting phenomena. The coexistence of both longitudinal and transverse fields can give rise to a circular polarisation in a plane transverse to the propagation, namely transverse SAM. The dual-symmetric SAM density of electromagnetic fields read[10], $\mathbf{S} = \mathbf{S}^e + \mathbf{S}^m = \frac{1}{4\omega}\Im\left\{\varepsilon_0 \mathbf{E}^* \times \mathbf{E} + \mu_0^{-1}\mathbf{B}^* \times \mathbf{B}\right\}$. In our beam setup, with the individual beam linear polarisation

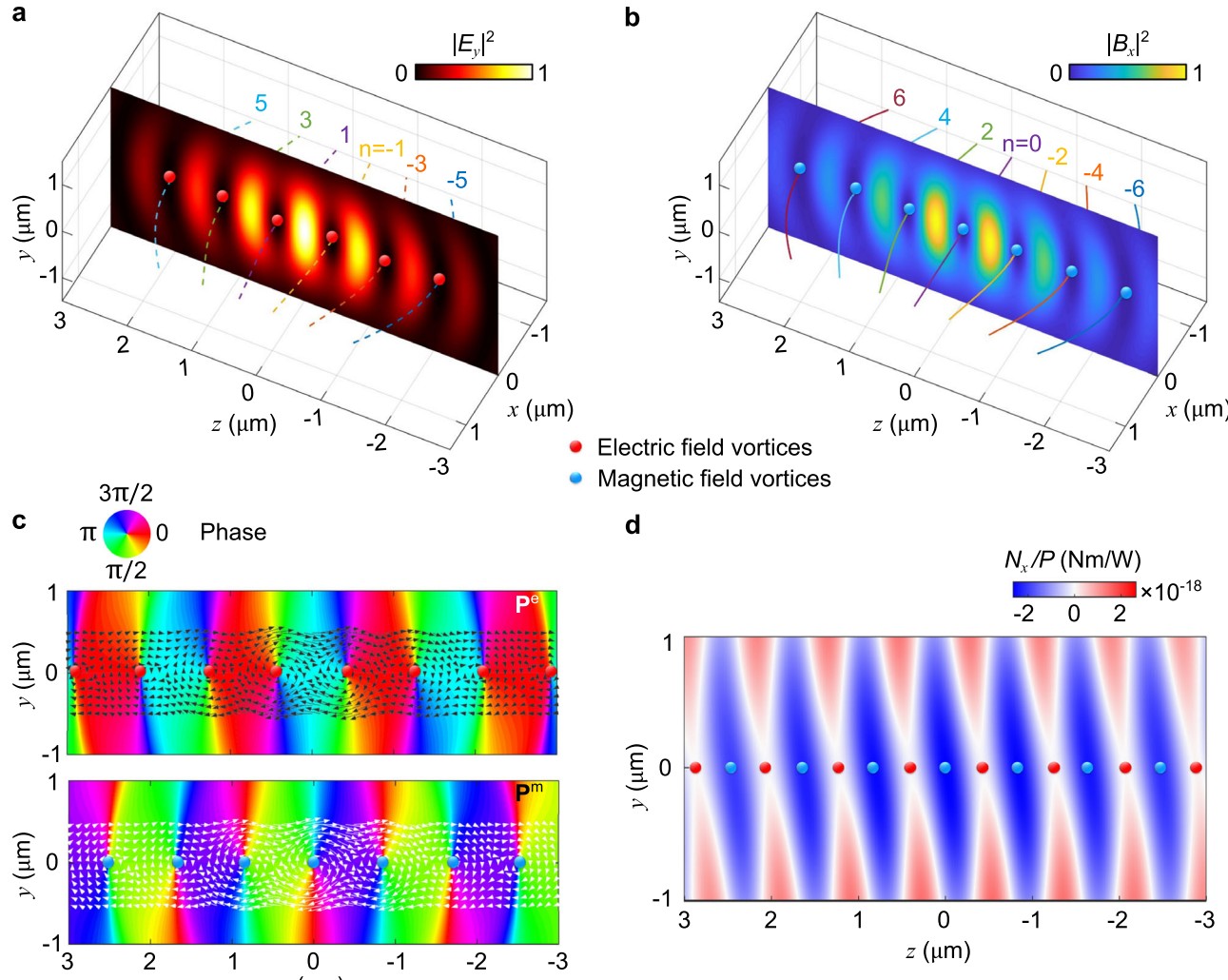

**Fig. 2 | Phase singularities and vortices due to electric and magnetic fields.** **a**, **b** Intensity profiles of the electric (**a**) and magnetic fields (**b**) in the axial plane. Dashed and solid curves show the phase singularity lines calculated via Eq. (7). **c** In-plane phase distribution for the transverse electric (upper) and magnetic (lower) fields. Arrows indicate the magnitude and direction of the electric and magnetic orbital momentum densities, $\mathbf{P}^e$ and $\mathbf{P}^m$, in the vicinity of singularity points. **d** Numerically calculated torque component $N_x$ acting on a $y$-oriented silicon nanorod placed at different positions in the $y$–$z$ plane, using Maxwell's stress tensor (see Methods). Each pixel comes from a full electromagnetic scattering simulation.

direction parallel to the direction of beam offset ($y$), the spin on the $x = 0$ plane is purely electric and has an $x$-component only, $S_x = S_x^e \propto \Im\left\{E_y^* E_z\right\}$. The calculated spin is shown in the Supplementary Information Section C, where we also visualise the polarisation state of the local electric field ellipse, with circular polarisations appearing at the field nodes. The spin exhibits a magnitude distribution similar to the electric field intensity, with a negative $x$-component of the spin near the intensity maxima.

**Transverse rotation measurements**

We probe the structured transverse optical momentum using a levitated optomechanical sensor. The force and torque acting on small dipolar scatterers can be directly associated with the field gradients and the local densities of optical linear and angular momenta in the illumination. In contrast, the optomechanical sensor used in our experiment is a silicon nanorod whose size is comparable to the optical wavelength, and its internal resonances can develop electric and magnetic high-order multipoles that interact with the incident fields. As a result, an intuitive description of the torque on the nanorod in this complex structured wavefield is not simple, and we resort to modelling

the torque using numerical techniques. Newton's second law of motion dictates that any change in linear/angular momentum invokes a force/torque. This principle leads to an induced torque on any particle within an optical field carrying OAM and SAM which we calculate using Maxwell's Stress Tensor, see Methods. The torque experienced by the nanorod is numerically computed for different positions of the nanorod on the $y$-$z$ plane, with the rod oriented along $y$, and the result is shown in Fig. 2d. The resulting torque is oriented in the $-x$ direction, as expected from the illuminating TOAM, and is strongest at the electric field maxima where the rod is levitated by optical gradient forces. With respect to a possible role of transverse SAM on the rotation, we numerically calculated the torque coming from the flux of the spin and the orbital parts of the angular momentum separately[37,38], showing that the SAM torque is negligible compared to the OAM torque (see Supplementary Information Section D).

Two linearly polarised (along the $y$ axis), counter-propagating laser beams are focused inside a vacuum chamber to form an optical trap (Fig. 3a), with the ability to introduce a separation $\delta_{\{x, y\}}$ in the $\{x, y\}$ directions respectively. We levitate silicon nanorods (Fig. 3c) and are able to track their motion in five degrees of freedom (Fig. 3b): three translational $\{x, y, z\}$ and two angular $\{\alpha, \beta\}$.

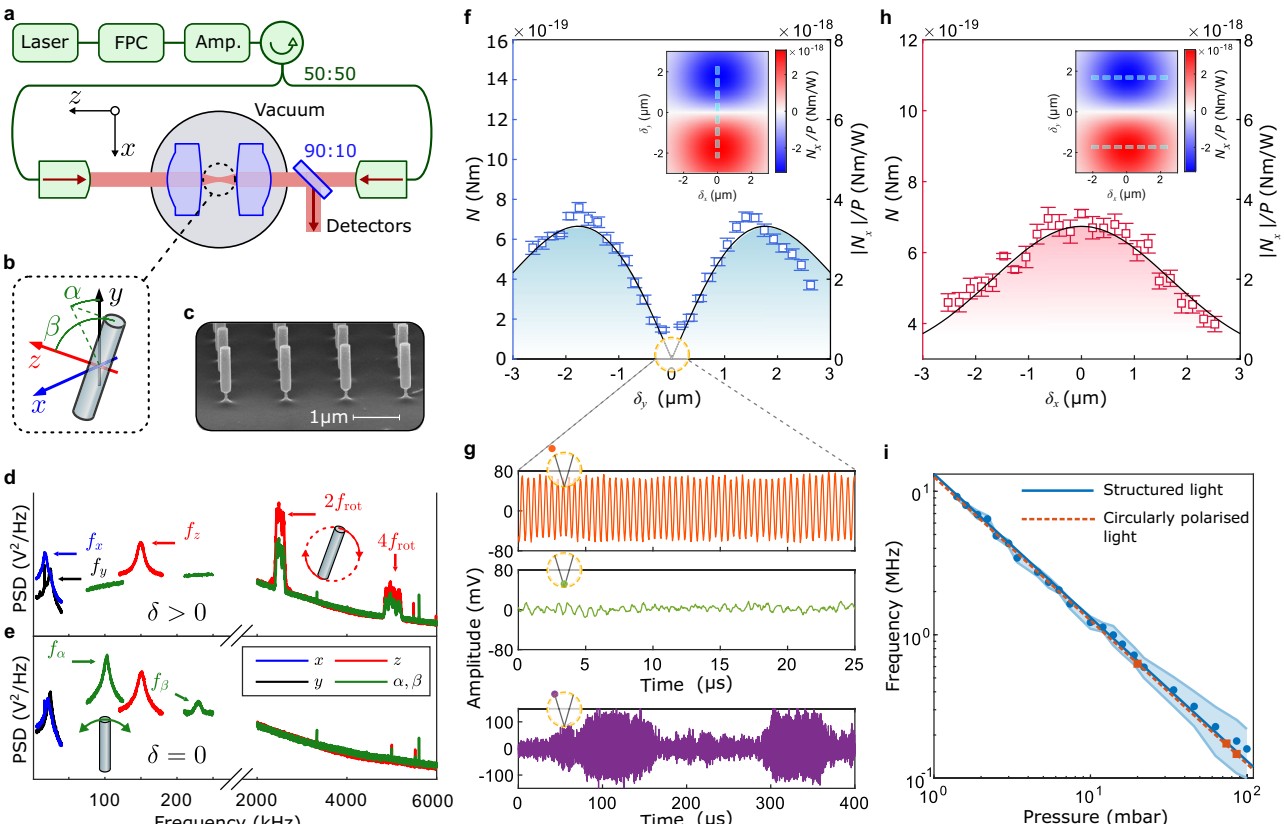

**Fig. 3 | Levitated optomechanical sensor. a** Light from a fibre-coupled laser system (1550 nm) with an laser amplifier (Amp.) and an integrated polarisation controller (FPC) is split into two equal arms. The light is out-coupled into free space, and focused through lenses (NA = 0.6) inside a vacuum system to create an optical trap. A separation between the beams is created by translating one of the fibre out-couplers (see Methods). **b** Coordinate axis for a nanorod trapped by linearly polarised light, undergoing harmonic motion in three linear axes $x$, $y$, $z$ and two librational axes $\alpha$, $\beta$. In this work, we drive rotations in the $y$–$z$ plane. **c** Scanning electron microscope image of the silicon nanorods used in this study. They are launched from a silicon substrate into the optical trap[52]. **d** Power spectral density (PSD) of the nanorod motion when driven to rotate at a frequency $f_{rot}$ by TOAM. The centre-of-mass is confined with harmonic frequencies $f_{\{x, y, z\}}$. **e** PSD of a nanorod with no TOAM present, showing that the particle no longer rotates and is additionally confined in the librational degrees-of-freedom $\{\alpha, \beta\}$. Vertical axis scales are not given as the data comes from a combination of detectors. **f** The

experimentally measured torque applied to the nanorod as a function of the offset $\delta_y$ (blue squares showing the mean value), compared to the theoretical prediction (solid black line). The inset shows the theoretical variation in applied torque with the 2D offset of the beams, and the dotted line indicated the separation explored in the figure. **g** Time-series of the signal (along the $z$ axis) for a nanorod undergoing rotation driven by TOAM (upper) for large $|\delta_y|$, and harmonically confined (middle) for small $|\delta_y|$ when the torque isn't large enough to drive rotation. At the boundary of these regimes, bistable rotations are observed (lower). **h** The effect of a transverse offset $\delta_x$ on the torque applied to the nanorod by TOAM (red squares showing the mean value), with comparisons to theory as in (**f**). **i** The rotational frequency scales linearly with pressure. Markers (blue circles) represent the mean value of $f_{rot}$ when driven by TOAM and the shaded areas represent the full range of $f_{rot}$. This can be compared with the rotational frequency of a nanorod driven by circularly polarised light (red squares).

When a separation $|\delta_y| > 0$ is introduced, intrinsic TOAM is generated at the optical antinode, where the nanorod is trapped. The optical OAM is transferred to the nanorod, driving it to rotate. This is evident in the frequency spectrum of the nanorod motion, where rotation at MHz rates $f_{rot}$ is detected (Fig. 3d), while the translational degrees-of-freedom remain harmonically confined with frequencies $f_{\{x, y, z\}}$. The $f_{rot}$ spectral feature is broad due to low-frequency drifts in the system[22] over the 10 s in which this data was acquired. When $\delta_x = \delta_y = 0$, there is no rotation, and the nanorod is harmonically confined in all five degrees-of-freedom (Fig. 3e). When rotating, there is no confinement in the $\beta$ direction, since this is the plane of rotation, and the motion in the $\alpha$ direction is gyroscopically stabilised[22].

The rotation rate of the nanorod is set by the balance between the optically induced torque $N$ and the damping due to the presence of gas $\Gamma$[22]:

$$f_{rot} = \frac{N}{2\pi I\Gamma}, \qquad (6)$$

where $I = \left(Ml^2\right)/12$ is the moment of inertia, $l$ is the nanorod's length, and $M$ its mass. Since $\Gamma$ is known[39], the rotation rate is a probe of the torque induced by TOAM. We compare to the theoretically calculated torque (see Methods) by measuring the variation in torque as $\delta_y$ is varied, and note a maximum in $N$ for a finite separation $|\delta_y| > 0$, in Fig. 3f, where the predicted torque is shown in the inset and by the solid line. If the rotation were simply due to the transfer of momentum from a single beam (e.g., in circularly polarised light) the maximum rotation rate would be when the optical intensity was a maximum ($\delta_y = 0$).

As $|\delta_y|$ is decreased, there is a transition from rotation (Fig. 3g, top panel) to harmonic confinement (middle panel) when the torque isn't large enough to overcome the optical potential which causes the nanorods to align along the polarisation direction. At the boundary, we observe bistable dynamics as stochastic forces due to collisions with gas molecules periodically drive the nanorod into rotation (lower panel). For finite $|\delta_y|$, a separation along the $x$ direction yields a single maximum at $|\delta_x| = 0$ (Fig. 3h), as predicted (solid line & inset).

We observe that the rotation frequency of the nanorod due to the TOAM depends linearly on the gas pressure (Fig. 3i, blue points), which

would not be the case for a motional frequency due to harmonic confinement[22]. Since the TOAM has a topological charge of −1, it would be expected that the optical torque is the same as for a particle exposed to circularly polarised light, which we confirm in Fig. 3i (red squares), where the separation is reduced to $\delta_{\{x,y\}} = 0$ and the polarisation of each beam is switched to circular. By comparing to the literature[25–27], we note that the maximal torque we induce on our silicon nanorods (~$3 \times 10^{-18}$ Nm) is five orders of magnitude larger than previously observed for particles levitated in vacuum. This is due to the shape-enhanced susceptibility of our nanorods[22] as compared to nanospheres, ellipsoids, or nano-dumbbells previously studied. Throughout this analysis, $\delta_z$ (i.e., the longitudinal separation of the two beam foci) is assumed to be zero. Supplementary Information Section E discusses the impact of $\delta_z$ on the optical torque.

## Discussion

In this work, we have presented a straightforward method for synthesising a robust and stationary array of optical vortices carrying intrinsic TOAM, and carried out a unique study of this exotic optical orbital angular momentum using a levitated optomechanical sensor as a probe. Being able to exert significant optical forces transverse to the propagation direction of free-space beams, without the need to have critical optical alignment, localised interference patterns, or polychromatic light, provides a powerful new tool for the optical manipulation of matter. Our work represents the first use of levitated particles as probes of structured light fields, exploiting the anisotropy of our nanorods to measure optically induced torque. By levitating the particle in a separate optical tweezers, the optical field could be mapped in three dimensions. The fact that the field structure can be sensitively detected by the particle probe opens the opportunity for the application of reactive quantities[40–43] in levitated optomechanics.

Furthermore, we have introduced an exciting new method for the precise optical control of nanoparticles levitated in vacuum[21], where the intrinsic nature of the TOAM and still-present optical trap at the electric field antinode enables the control of alignment and rotation without driving orbits. This new way of manipulating nanoparticles will be instrumental in cavity optomechanics[34] and the quantum control and exploitation of rotation[31]. The large optical torque we exert on the levitated silicon nanorods will enable sensitive torque sensing[26], and the transverse direction of the particle's angular momentum will allow the rotating particle to be brought close to surfaces to measure quantum friction[44], lateral Casimir[45], and other short-range forces[46].

## Methods

### Phase structure evaluation

The phase singularities arising in the electric and magnetic fields of Eq. (1) can be analytically found. In fact, the phases of the electric and magnetic field, $\phi_e$ and $\phi_m$, can be written as

$$\tan\phi_e = \frac{\Im\{E_y\}}{\Re\{E_y\}} = \frac{C(x,y-\delta,z)-C(x,y+\delta,z)}{D(x,y-\delta,z)+D(x,y+\delta,z)},$$
$$\tan\phi_m = \frac{\Im\{B_x\}}{\Re\{B_x\}} = \frac{C(x,y-\delta,z)+C(x,y+\delta,z)}{D(x,y-\delta,z)-D(x,y+\delta,z)}, \tag{7}$$

where

$$C(x,y,z) = e^{\frac{-kr^2 z_0}{z^2+z_0^2}}\sin\left(\frac{kr^2 z}{z^2+z_0^2} - \varphi(z) + kz\right),$$
$$D(x,y,z) = e^{\frac{-kr^2 z_0}{z^2+z_0^2}}\cos\left(\frac{kr^2 z}{z^2+z_0^2} - \varphi(z) + kz\right), \tag{8}$$

with $r^2 = x^2 + y^2$.

For $\delta = 0$, $\phi_e$ and $\phi_m$ are constant, and thus the phase profile will be planar. This is the expected result for a standing wave. The appearance of a phase singularity or dislocation, which is the physical

origin of phase vortices, requires $\delta \neq 0$, and

$$\Im\{E_y\} = \Re\{E_y\} = 0 \tag{9}$$

for electric field singularities, or

$$\Im\{B_x\} = \Re\{B_x\} = 0 \tag{10}$$

for magnetic field singularities[36]. The intersection lines of $y = 0$ and Eq. (3) are the solutions to Eq. (9) for odd $n$ and to Eq. (10) when $n$ is even.

### Numerical simulations and calculations

A particle can experience optical forces by either absorbing or scattering light, as accounted for by the Maxwell stress tensor (MST) $\overleftrightarrow{\mathbf{T}}$ which represents the overall time-averaged flow of momentum in an electromagnetic field[42,47,48],

$$\overleftrightarrow{\mathbf{T}} = \frac{1}{2}\Re\left\{\varepsilon_0 \mathbf{E}\otimes\mathbf{E}^* + \mu_0^{-1}\mathbf{B}\otimes\mathbf{B}^* - \frac{1}{2}(\varepsilon_0|\mathbf{E}|^2 + \mu_0^{-1}|\mathbf{B}|^2)\overleftrightarrow{\mathbf{I}}\right\}, \tag{11}$$

where $\otimes$ corresponds to an outer product, and $\mathbf{I}$ is the three-dimensional (3D) identity matrix. This can be extended to electromagnetic angular momentum via a cross product with the spatial coordinates $\mathbf{r}\times\overleftrightarrow{\mathbf{T}}$, and the torque is calculated from the surface integral of a closed surface enclosing the particle[37,38,49],

$$\mathbf{N} = \oiint (\mathbf{r}\times\overleftrightarrow{\mathbf{T}})\cdot\hat{\mathbf{n}}\,\mathrm{d}S, \tag{12}$$

where $\mathbf{N}$ is the torque, $\mathbf{r}$ is the position vector, and $\hat{\mathbf{n}}$ is the unit vector normal to the surface $S$.

A calculation of the torque using the MST is a time-consuming process that requires knowing the total electric and magnetic fields incident on, and scattered by, the nanorod. These fields are calculated numerically as described below. The nanorod was approximated as a cylinder, and the relative permittivity of silicon was assumed to be 12.1.

For Fig. 2d, the 3D electric and magnetic fields scattering off the nanorod are required for each pixel in the colourplot, with the nanorod placed at different positions within the structured illumination. In order to do this efficiently, a procedure previously implemented in ref. 50 was followed, in which each beam is decomposed into a collection of plane waves using a spatial Fourier transform. To incorporate the nanomechanical sensor, each plane wave component is then replaced with a numerical simulation of the total fields from a nanorod illuminated with the same plane wave. The nanorod's cylindrical symmetry is exploited to reduce the number of unique simulations.

The beam is then reconstructed using an inverse Fourier transform and the beam is augmented with the appropriate total scattering of the nanorod[51]. In this way, the only required numerical simulations are those of the nanorod under plane wave illumination in vacuum at various angles of incidence, easily performed via frequency-domain 3D finite-element-method simulations using the commercial software package *CST Microwave Studio* (Dassault Systemes). Thanks to the linearity of Maxwell's equations, the fields for individual plane waves can be combined, with appropriate weighted amplitudes and phases, to synthesise the scattering from any desired structured far-field illumination–a step done in post-processing using Mathworks's numerical computation software *MATLAB*–which was applied to synthesise the counter-propagating Gaussian beams. The phases of the plane waves can be adjusted to shift the position of the beams and hence sweep the different nanorod locations. For each location, the end result of this post-processing is the full 3D scattering field of the nanorod in a given position of the optical trap to no approximation beyond numerical accuracy.

Once the 3D fields are obtained, calculating the torque requires performing the integration of Eq. (12) over an arbitrary surface enclosing the nanorod. We chose a cube centred around the nanorod, and varied the cube's dimensions to ensure convergence of the result.

The theoretical torque lines and insets of Fig. 3f, h are also calculated via this numerical simulation method.

## Experimental system

We use a standing wave optical dipole trap formed by two counter-propagating linearly polarised light beams focused inside a vacuum chamber. As illustrated in Fig. 3a, a pair of 1550-nm Gaussian beams (~0.6 W) are focused by two NA = 0.6 lenses (clear aperture 3.60 mm, effective focal length 2.97 mm, Thorlabs 355660-C), such that their foci coincide, and a silicon nanorod is levitated in the antinodes of the resultant standing wavefield. The nanofabricated silicon nanorods (880 nm length and 210 nm diameter, Kelvin Nanotechnology) are grown on a silicon wafer (Fig. 3c) before directly launching to the optical trap by Laser-Induced Acoustic Desorption (LIAD)[52] at a pressure of a few millibar. At this point they are deeply and stably confined within one optical antinode of the standing wave trap.

Although the optical field defines where the particles are trapped in the $x$–$y$ plane, they can be trapped in one of several antinodes of the optical standing wave. The field intensity, and therefore the torque, varies at the different antinodes. Our experimental measurement provides the absolute value of the torque, while our calculations provides the torque at the central antinode normalised by the power carried by the beams. This explains the dual vertical axis in Fig. 3f, h. The nanorods can be translated in the $z$-direction by changing the relative phase of the two beams, which we achieve by translating one of the fibre out-couplers in the $z$-direction. The relative phase between the two beams is monitored, but not stabilised, and only has significant noise below 100 Hz, far from any resonance of our trapped particles.

To generate the offset between the two beams, we linearly translate one of the fiber-out-couplers by an amount $\Delta$. The two beams still overlap at the focus, but away from the focus an offset $\delta$ is introduced. The translation $\Delta$ and the offset $\delta$ are linearly proportional to each other[53], with a proportionality constant that depends on the position along the $z$ axis. Since we do not know the absolute position along the $z$ axis, we treat this proportionality constant as a free parameter, when comparing theory to data in Fig. 3f, h. This described technique for generating a separation also induces a small angle between the beams (~0.04 rad), which has a negligible effect on the results of our numerical simulations (see Supplementary Information Section F).

## Detection methods

We detect the motion of the levitated nanorods using a variety of methods, all based on collecting the light which has passed through the focus of the optical trap (Fig. 3a). More details can be found in ref. 22. The $\{x, y\}$ motion is measured by imaging the beam onto a quadrant photodiode. The $z$-motion is monitored using balanced homodyne with light that hasn't interacted with the particle. This method is sensitive to rotation in the $y$–$z$ plane, since the intensity of the collected scattered light is strongly dependent on the alignment of the nanorod (Fig. 3d). The $\{\alpha, \beta\}$ motion is detected by passing the light through a polarising beam-splitter and performing a balanced detection on the two output ports.

## Data availability

Data that support the plots within this paper and other findings of this study are available from the corresponding author upon request.

## Code availability

The code used in the present work is available from the corresponding authors upon request.

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

## Acknowledgements

We thank Dr. Steven Simpson for useful early discussions. XX acknowledges the National Natural Science Foundation of China (12274181, 11804119), and Guangdong Basic and Applied Basic Research Foundation (2023A1515030143). J.M. recognises support from the European Research Council Grant Agreement Nos. 803277 & 957463 and Royal Society Research Grant RGS\R1\201096. J.J.K.S. and F.J.R.F. recognise support from the European Research Council Starting Grant No. ERC-2016-STG-714151-PSINFONI.

## Author contributions

Y.H. built the experiment, took and analysed data, performed numerical simulations and contributed to the manuscript. M.N. assisted with taking and analysing data, and preparing the manuscript. J.A.S. assisted with taking data. J.J.K.S., F.J.R.F. & X.X. constructed theoretical models, performed simulations and contributed to the manuscript. In addition, X.X. first described the underlying physical process. J.M. assisted in data analysis, contributed to the manuscript and conceived of the experiment.

## Competing interests

The authors declare no competing interests.
