## [Peer Review File · Nature Communications]

Structured transverse orbital angular momentum probed by a levitated optomechanical sensorREVIEWER COMMENTS

Reviewer #2 (Remarks to the Author):

This is a well written paper describing new results on the experimental observation of the effects of transverse orbital angular momentum on a nano-rod optically trapped by two tightly-focused counter-propagating parallel but offset beams with identical linear polarization. The observed torque is orders of magnitude stronger than that observed in previous groups, with ~MHz rotation speeds achieved at ~mbar pressures, in part due to the geometric structure of the nanorod. The authors make a convincing case that what they observe is actually rotation from TAOM and not due to spin angular momentum e.g. due to birefringence in the material and imperfect laser polarization. The work is novel and will be a useful addition to the field, potentially enabling applications in torque sensing and rotational sensing, including possibly investigating effects such as Casimir effects and quantum friction, and I recommend publication. I have a couple of minor comments on Fig. 3 manuscript.

1. The horizontal axis for Fig.3 (d,e) is labeled in units of Hz. I assume this is a typographical error and this should be kHz?
2. The lowest tick marker on the vertical axis of Fig. 3(i) is labelled as 10^1 MHz. I assume this should be 10^{-1} MHz.

The structured light fields have been the focus in the study field of optical manipulation due to many interesting phenomena and important applications. In this manuscript, the authors present a study of detecting a structured transverse orbital angular momentum of the lights using a nanorod trapped by these beams. The transverse orbital angular momentum (TOAM) is produced by two parallel and counter-propagating linearly-polarized focused beams when they are slightly separated. Therefore, a nanorod trapped by these beams, as a high-sensitive sensor, can feel the torque and is driven to rotate. In this work, the authors try to use the mechanics of transverse orbital angular momentum of light to explain generating the transverse rotation of nanoparticles. From my point of view, the trapping laser generates a forward scattering force for the levitated nanoparticle due to optical scattering, and therefore, two parallel and counter-propagating linearly-polarized focused beams with slight separation will generate a torque on a nanorod as shown below figure. From this perspective, two counter-propagating linearly-polarized focused beams may have different frequencies, and therefore do not generate interference and the transverse orbital angular momentum, which still can create the torque on the nanorod. So I do not think the transverse orbital angular momentum directly relates to the transverse rotation of the nanorod. Thus I do not recommend this work on NC in the present form.

Moreover, I have comments and this work.

detailed questions on

1. The radial and angle separation are considered. In fact, the longitudinal separation along z axis (the two focus planes of the two beams do not coincide) is inevitable in the experiment. In this case, the phases and the intensity distributions of the electric and magnetic fields will be changed when the longitudinal separation is not zero. Therefore, an explanation about the influence or why it is neglected is necessary.
2. The two counter-propagating beams form a standing wave, in some cases, the trapped particles can jump between the adjacent antinodes, which does not be mentioned in this paper. Does it not influence the measurement?

3. The interference of the two beams is much sensitive to the stability of the system, which can be easily disturbed by the external environment, what kind of stabilization method is utilized in the experiment?

4. The important experimental parameters, such as the power of the trapping beams, the optical aperture and the focal length of the high NA lenses, are missed, which are much useful for the readers to estimate the experimental results.

5. The signal shapes of the rotation in Fig.3.d is roughly rectangular, which is much different with the signal of a common optically-levitated rotor in vacuum (it is usually a Lorentzian function). Are there any other effects? it is better for the author to provide an explanation. In addition, from the usual point of view considering the focused beam, f_{β} is smaller than f_{α} , however, in Fig.3.e, f_{α} is larger than f_{β} , can the authors present an explanation?

6. Some other important works of the longitudinal SAM are missed, for instance:

Reimann, R. et al. GHz rotation of an optically trapped nanoparticle in vacuum. Phys. Rev. Lett. 121, 033602 (2018).

Ahn, J. et al. Optically levitated nanodumbbell torsion balance and GHz nanomechanical rotor. Phys. Rev. Lett. 121, 033603 (2018).

Answer to reviewer's comments Manuscript NCOMMS-22-44541-T

We sincerely thank the reviewers for their reviews and comments, which have helped us in improving our manuscript and in particular encouraged us to clarify more effectively some key points.

Below we answer the individual reviewers' points.

1 REVIEWER 1

Reviewer 1 states:

"The structured light fields have been the focus in the study field of optical manipulation due to many interesting phenomena and important applications. In this manuscript, the authors present a study of detecting a structured transverse orbital angular momentum of the lights using a nanorod trapped by these beams. The transverse orbital angular momentum (TOAM) is produced by two parallel and counterpropagating linearly-polarized focused beams when they are slightly separated. Therefore, a nanorod trapped by these beams, as a high sensitive sensor, can feel the torque and is driven to rotate. In this work, the authors try to use the mechanics of transverse orbital angular momentum of light to explain generating the transverse rotation of nanoparticles. From my point of view, the trapping laser generates a forward scattering force for the levitated nanoparticle due to optical scattering, and therefore, two parallel and counterpropagating linearly-polarized focused beams with slight separation will generate a torque on a nanorod as shown below figure. From this perspective, two counter-propagating linearly-polarized focused beams may have different frequencies, and therefore do not generate interference and the transverse orbital angular momentum, which still can create the torque on the nanorod. So I do not think the transverse orbital angular momentum directly relates to the transverse rotation of the nanorod. Thus I do not recommend this work on NC in the present form."

Our answer:

We thank the referee for his/her comments. We agree completely with the referee in the interpretation for the torque based on the two beams "pushing" via a scattering force (a.k.a., radiation pressure) the two ends of the nanorod in opposite direction, as clearly depicted in his/her figure.

In our reply, we hope to convince the reviewer that this "out-of-centre pushing force" described in their figure is *precisely* the very definition of orbital angular momentum, and therefore the referee's description is, in fact, exactly equivalent to a description in terms of transverse orbital angular momentum which we present in the manuscript. We will also address, at the end, the very interesting comment about the beams having different frequencies.

As a first step, we would like to point the reviewer to the momentum density vectors in the illumination (\mathbf{P}^e and \mathbf{P}^m) whose arrows are drawn in our Fig. 2(c). These momentum densities represent the linear momentum of light in the illuminating beams, and they are in effect exactly equivalent to the big red arrows in the reviewer's sketch. In our figure 2(c), as also happens in the

reviewer's sketch, linear momentum is pointing right for the top beam, and is pointing left for the bottom beam.

Indeed, this momentum density will exert a radiation pressure on each part of the rod. Let's consider that our rod is made up of a collection of differential volumes. Under a crude approximation (used only to illustrate this answer to the reviewer) we could simplify the situation and assume that each volume element of the rod is subject *only* to the illuminating field (i.e., we neglect the scattering from one part of the rod acting on another part). The *pushing force* or *scattering force* acting on each volume element \mathbf{f}_{SF} would then be proportional to the local momentum density vector of the illumination at each volume element $\mathbf{f}_{SF} \propto \mathbf{P}$ where $\mathbf{P} = \mathbf{P}^e + \mathbf{P}^m$. According to this approximation, the net force on the rod would be $\mathbf{F} = \iiint_V \mathbf{f}_{SF} dV \propto \iiint_V \mathbf{P} dV$ which in this case would evidently be zero (due to the symmetry of the illumination pushing the top and bottom parts in opposite directions).

However, these balanced forces acting on each part of the volume will clearly exert a non-zero *net torque* on the rod, initiating a rotation, essentially by pushing the top end to the right, and the bottom end to the left. The mechanical torque caused by a force on an object, evaluated around the origin of coordinates, is defined as $\boldsymbol{\tau} = \mathbf{r} \times \mathbf{f}$. The *net torque*, therefore, by integration of the torque on each volume element, is given by:

$$\mathbf{N} = \iiint_V \mathbf{r} \times \mathbf{f} dV \quad (R1)$$

If we again make the above assumption that each part of the rod only feels the illuminating field, then $\mathbf{f}_{SF} \propto \mathbf{P}$, and so the torque can be written as:

$$\mathbf{N} \propto \iiint_V \underbrace{\mathbf{r} \times \mathbf{P}}_{\mathbf{L}} dV \quad (R2)$$

where the term $\mathbf{L}(\mathbf{r}) \equiv \mathbf{r} \times \mathbf{P}(\mathbf{r})$ is, in fact, the *definition* (as described in our paper) of the orbital angular momentum density of light.

In other words, the two red arrows in the reviewer's diagram, which represent the beam's linear orbital momentum \mathbf{P} , are also associated with an OAM, $\mathbf{r} \times \mathbf{P}$ which we add below as a green arrow to the reviewer's sketch. These green arrows point, for both beams, in the direction into the page, i.e. it is a *transverse orbital angular momentum*. Although red and green arrows are shown only as a single arrow, to be technically precise we should draw a small arrow (a vector *field*) defined at every point in space $\mathbf{L}(\mathbf{r}) = \mathbf{r} \times \mathbf{P}(\mathbf{r})$ representing the OAM density of the illumination, which can then be integrated over the volume of the rod to obtain the net torque.

This simple diagram shows why, by virtue of being misaligned with respect to the origin, each of the two beams creates a transverse orbital angular momentum (OAM) density at the volume occupied by the rod. **This, being our explanation of the torque, is indeed equivalent (and in complete agreement) to the reviewer’s intuition.**

The fact that the reviewer saw their explanation as if it was opposed to our description suggests that we did not do a good job in explaining, intuitively, the physics of the transverse OAM, and so we have made changes to the manuscript to make this interpretation, and the equivalence, as clear as possible. This is added in the paragraph after Eq. (4), and a new supplementary section is added.

The existence of TOAM in this illumination is simple to understand if we imagine an extended particle, such as our rod, placed at the origin (Fig. 1): roughly speaking, opposite ends of the rod will be pushed (via an optical pressure force caused by each beam’s orbital momentum density \mathbf{P}) in opposite directions, generating a torque on the particle. This torque is therefore enabled by the transverse nature of the OAM density $\mathbf{L} = \mathbf{r} \times \mathbf{P}$ (see Supplementary Information for details).

We would also like to clarify a possible misconception between OAM and vortices, which might have given rise to the reviewer’s initial rejection of our claim that OAM is the source of the torque. The existence of OAM is not necessarily linked to the existence of a vortex. In fact, in the paper, we *do not* claim that the vortices are causing the rotation: we only say that the OAM is causing the rotation. The array of vortices are presented as an extremely interesting and important feature of this setup, but not as the cause of the torque. Please note that a vortex is simply a situation in which the momentum density \mathbf{P} circles around a point, and this results in the total integrated angular momentum $\langle \mathbf{L} \rangle = \iiint_{-\infty}^{\infty} \mathbf{r} \times \mathbf{P} dV$ having an *intrinsic* term, meaning that it is a term independent of the origin’s location. In general, changing the origin of coordinates from 0 to \mathbf{r}_0 implies a change in integrated angular momentum $\langle \mathbf{L} \rangle \rightarrow \langle \mathbf{L} \rangle + \mathbf{r}_0 \times \langle \mathbf{P} \rangle$, as explained in the paper. This means that the OAM of a single misaligned linearly polarized beam can be made to go to zero if we move the origin to a suitable location (along the axis of the beam). In contrast, the OAM of a vortex cannot be made to go to zero, it is an *intrinsic* OAM. Here, we have two linearly polarized beams with a certain offset. Each of them have an *extrinsic* OAM: meaning that their OAM can be made individually zero by displacing the origin. However, taken together, because they are displaced in different directions, there is no location of the origin that would result in zero OAM – and therefore the resulting OAM is *intrinsic* even if there had been no vortices involved.

Next, before addressing the comment about the different frequencies, we would like to comment on the approximation made earlier. The approximation was great for gaining an intuitive understanding of the situation, but in fact it is a very crude and unsuitable approximation. This rod is big (when compared against the wavelength) and it interacts strongly with the illumination. The strong scattering field will greatly modify the net electromagnetic fields around and inside the rod. This means that the scattering from one part greatly affects the force experienced by another part, and this is why the torque on the rod cannot be simply calculated as a volume integral of the OAM (unfortunately for us, as this would have made the calculations *and explanations* much easier).

To take the full physics into account, we applied the Maxwell Stress Tensor formalism. This formalism calculates the torque from first principles, via conservation of angular momentum, and uses the full scattered fields. Upon conducting this numerical and computationally intensive calculation, we arrived at the transverse torque map shown in Fig. 2(d). In the above discussion we also ignored the fact that light’s spin can also exert a torque on each volume element. When writing the paper we were concerned by this, and that is why we decided to split Maxwell’s stress tensor into both spin and orbit

angular momentum components, which is a novel procedure introduced theoretically in Ref. [39] and applied, to our knowledge, for the first time in this work. The results show that the transverse torque is dominated by the orbital, and not the spin component, of angular momentum (cf. Fig. S4). Thus we confidently attribute the rotation to the OAM – and the OAM ultimately stems from the misaligned beams pushing the opposite ends of the rod in opposite directions.

Finally, we address the interesting remark about beams with different frequencies. Here, we disagree with the reviewer’s statement “*two counter-propagating linearly-polarized focused beams may have different frequencies, and therefore do not generate interference and the transverse orbital angular momentum*”.

Even if the two beams have different frequencies, light’s linear orbital momentum density can still be defined (but it is now a time-dependent quantity $\mathcal{P}(\mathbf{r}, t)$, so we cannot work with complex phasors), and it is still associated with a time-dependent orbital angular momentum density $\mathcal{L}(\mathbf{r}, t) = \mathbf{r} \times \mathcal{P}(\mathbf{r}, t)$. In fact, the case of the beams having two different frequencies is a situation that we were studying before receiving this review, and we have interesting results with hopes of publishing a future work -and which therefore we would prefer not to discuss in the present version of the manuscript. We present here some preliminary results to the reviewer showing that, in fact, when two such beams with different frequency are used, the interesting transverse OAM structure is *still there*, including the array of vortices! But this time, the array of vortices move in space as time progresses (See Supplementary Video R1, only included with this response letter). After discussion with experts on the topic, these moving vortices represent “spatio-temporal vortices” which have been rising in popularity recently, and thus we are pursuing further study and a publication about this.

To conclude, the argument of two beams with different frequencies does not contradict the fact that the rotation of the rod is caused by the OAM structure of the illumination. And this OAM structure is caused -by definition- by the fact that the two beams are misaligned with respect to the center, which was the reviewer’s correct intuition all along.

The new supplementary section summarizes, in more technical language, the explanations above:

Simplified model for the origin of TOAM and torque:

Consider the rod under illumination by the two beams (Fig. 1). As we have shown in the main text, a TOAM density is present in this illumination. This TOAM can be understood in very simple terms: opposite ends of the rod will be pushed (via an optical pressure force caused by each beam’s orbital momentum density \mathbf{P}) in opposite directions, generating a torque on the particle.

We will simplify the situation with an approximate model by assuming that each volume element of the rod is subject to the illuminating field *only* (i.e., this model neglects the scattering from one part of the rod acting on another part). The *scattering force* acting on each volume element \mathbf{f}_{SF} would then be proportional to the local momentum density vector of the illumination at each volume element $\mathbf{f}_{SF} \propto \mathbf{P}$ where $\mathbf{P} = \mathbf{P}^e + \mathbf{P}^m$. According to this approximation, the net force on the rod would be $\mathbf{F} = \iiint_V \mathbf{f}_{SF} dV \propto \iiint_V \mathbf{P} dV$ which in this case would be zero due to the symmetry of the illumination pushing the top and bottom parts in opposite directions.

However, these balanced forces will exert a non-zero *net torque* on the rod, essentially by pushing the top and bottom of the rod in opposite directions. The mechanical torque caused

by a force on an object, evaluated around the origin of coordinates, is defined as $\boldsymbol{\tau} = \mathbf{r} \times \mathbf{f}$. The *net* torque, therefore, by integration of the torque on each volume element, and applying the same model above where $\mathbf{f}_{SF} \propto \mathbf{P}$, is given by:

$$\mathbf{N}_{\text{model}} = \iiint_V \mathbf{r} \times \mathbf{f} \, dV \propto \iiint_V \underbrace{\mathbf{r} \times \mathbf{P}}_{\mathbf{L}} \, dV$$

where the term $\mathbf{L}(\mathbf{r}) \equiv \mathbf{r} \times \mathbf{P}(\mathbf{r})$ is the *definition* of orbital angular momentum density of light, which in this case becomes transverse. By virtue of being misaligned with respect to the origin, each of the two beams creates a TOAM density at the volume occupied by the rod which can be integrated to estimate the direction of the torque. Interestingly, this explanation does not even require the presence of the transverse vortex arrays. In this model we also ignored the SAM density of light which can also cause a torque on the individual volume elements.

This model is great for gaining an intuitive understanding of the torque, but it is unsuitably crude when quantitatively assessing the torque. On one side, SAM should be included. On the other hand, the rod is big compared to the wavelength and it interacts strongly with the illumination. The scattering from one part greatly affects the force experienced by another part, and so the torque on the rod cannot be simply calculated as a volume integral of the OAM and SAM.

To take the full physics into account, in the main text we applied the Maxwell Stress Tensor formalism to calculate the torque from first principles, via conservation of angular momentum, using the full scattered fields from numerical simulations (Fig. 2d).

Reviewer 1 states:

“Moreover, I have detailed comments and questions on this work.

1. The radial and angle separation are considered. In fact, the longitudinal separation along z axis (the two focus planes of the two beams do not coincide) is inevitable in the experiment. In this case, the phases and the intensity distributions of the electric and magnetic fields will be changed when the longitudinal separation is not zero. Therefore, an explanation about the influence or why it is neglected is necessary.”

Our answer:

The reviewer makes a good point that the longitudinal separation of the beams’ foci (δ_z) was not discussed in the paper. Indeed experimentally, two counter-propagating beams will never be perfectly confocal in the z -axis to infinite precision ($\delta_z \neq 0$). This was originally neglected in the paper for brevity. However, we have conducted some further analysis to satisfy the reviewer and have ultimately agreed that this information should be added to the revised manuscript. A new section in the Supplementary Information (now labelled D) has been created to encompass the following information.

We can also model the δ_z dependence of the optical torque on a particle at the centre of the interference pattern by using the same Fourier transform procedure mentioned in the Methods section of the paper. Further information of this procedure has also been provided via an additional reference. The aim is to show the optical torque on the silicon nanorod from the paper fixed at the origin and orientated along the y -axis. The particle is illuminated with one beam propagating along

$-z$ and focused at $(0, \delta_y, \delta_z)$, and another beam propagating along $+z$ and focused at $(0, -\delta_y, -\delta_z)$. Both beams have a wavelength of 1550nm, a beam waist of 1600nm and are linearly polarised along y . For these calculations, δ_y is fixed at 500nm.

Figure R2: The evolution of the free-space transverse vortices field as the foci of the two beams are separated longitudinally by $2\delta_z$.

We observe the periodic transverse vortices when $\delta_z = 0$ as expected, as was shown in the main paper. Once $\delta_z \neq 0$, the vortices begin to gradually breakdown and transition into a regular standing wave pattern with no appreciable variation of phase over y . However, this breakdown occurs gradually over a distance roughly one order of magnitude larger than the wavelength. A more granular depiction of this transition from vortices to regular standing wave behaviour is shown in the attached video, where δ_z is varied over time and the corresponding field properties are shown.

Experimentally, the alignment of the foci along z is achieved by ensuring that the outward propagating beams remains collimated over a course of several metres. In other words, we minimise δ_z by maximising the collimation of the beam after it has passed through the two confocal lenses. We can also approximate the experimental δ_z by observing the amount of beam divergence and using ray optics. As a rough estimate, the beam diameter doubled after propagating 5 metres from the second lens, which has a focal length of 2.97mm and a lens diameter of 4mm. Using the thin lens approximation and the imaging equation, we obtain an order-of-magnitude estimate of $\delta_z \approx 2\mu\text{m}$. We can then infer from this and Fig. R2 that the transverse vortices are present in the experiment.

Simulations of the optical torque can also be conducted using these incident fields. By inserting the particle at the origin and using the Maxwell stress tensor approach, we can calculate the δ_z dependence of N_x . Figure R3 depicts the results of these simulations and indicates that the transverse torque is maximal when the two beams share approximately the same focal plane (any deviation from $\delta_z = 0$ is hypothesised to be down to numerical errors in simulations). When the beams are separated along z , the particle no longer absorbs TOAM from the incident fields and the torque diminishes. Noise in this data is associated with numerical noise from the CST Microwave Studio plane wave simulations.

When combining this with the experimental approximation for δ_z and simulated results in Fig. 3f,h in the paper, we believe this provides strong evidence that we have indeed measured the proposed system, and that δ_z is suitably small in the experimental measurements.

Figure R3: The transverse torque on the silicon nanorod as a function of longitudinal focal plane separation. The wavelength is $1.55\mu\text{m}$.

The new Supplementary Information section reads:

Throughout the main paper (excepting Fig. 3h), the relative position of the two counter-propagating beams is defined by restraining $\delta_x = \delta_z = 0$ and varying δ_y . The δ_z separation can be difficult to quantify precisely in some experimental configurations so it can be instructive to investigate the δ_z dependence of the transverse optical torque in this optical trap, and what limitations it may present.

The intensity and phase of the electric field in the counter-propagating trap with $\delta_y = 500$ nm (roughly $\lambda/3$) is presented in Fig. S3a-e but with different values for δ_z . As δ_z increases, the transverse phase vortices transition into regular standing wave mode structures with no significant change of phase along the y direction. This suggests that the TOAM present in the optical field is diminishing once the focal planes of the two beams are significantly far apart longitudinally. Such a result is expected given the nature of tightly focused beams and the rapid change in their fields as one moves away from the focus.

Building on these results, the δ_z dependence of the optical torque on the silicon nanorod probe can be simulated by inserting the particle's scattering into the incident fields and then using the Maxwell stress tensor approach. The torque is of greatest magnitude when $\delta_z = 0$ and the transverse vortices are most prevalent. Once the distance between the focal planes of the two beams ($2\delta_z$) is roughly an order of magnitude greater than the wavelength, the torque greatly diminishes. We also observe some mild asymmetry in the particle's response with respect to δ_z . The size of this constraint will in general depend on the degree of beam focusing and the specifics of the trapping system. An order-of-magnitude estimation for our experimental δ_z can be generated by noting the beam divergence after the trapping lenses (beam diameter doubles after 5 m), the lens focal length (2.97 mm) and the lens

diameter (4 mm). Simple ray optics can be used to find an experimental estimation of $\delta_z \approx 2 \mu\text{m}$. We therefore conclude that our trapping system has a suitably small foci separation for generating an incident field with transverse phase vortices and generating a torque via TOAM.

Reviewer 1 states:

“2. The two counter-propagating beams form a standing wave, in some cases, the trapped particles can jump between the adjacent antinodes, which does not be mentioned in this paper. Does it not influence the measurement?”

Our answer:

We agree that in experiments where particles are levitated in counterpropagating standing-wave optical tweezers, it is possible to observe hopping between optical antinodes, indeed we observe this when working with optically levitated nanospheres. The effect is very obvious via all of our detection methods, due to the motion of the particle over an optical node leading to a minimum in light scattering.

We do not observe this phenomenon in the experiments presented in this manuscript, the reason being that our silicon nanorods have a significantly higher susceptibility, and hence experience a deeper optical potential than the silica nanospheres typically used in the literature. In our particular case, the optical potential for our nanorods is about 5 times deeper than for a silica sphere of the same volume. At the relatively high optical powers (0.6W) and typical beam-waist ($1\mu\text{m}$) in our experiment, the maximal optical potential depth is significantly greater than 1 million Kelvin, giving a quantitative reason for the particle stability.

We have added “At this point they are deeply and stably confined within one optical antinode of the standing wave trap.” to the **Experimental system** section of the Methods.

Reviewer 1 states:

“3. The interference of the two beams is much sensitive to the stability of the system, which can be easily disturbed by the external environment, what kind of stabilization method is utilized in the experiment?”

Our answer:

Here we assume the referee is referring to the relative phase stability of the two counterpropagating beams. This is something which we are mindful of. Our stabilization is entirely passive: the majority of our optical path is within thin-jacket optical fibre, securely taped to an optical table, in a room which is temperature stabilized to better than 1 degree centigrade, with the fibres not exposed to air flow. The optical table is passively vibration isolated. We monitor the relative phase shift on a balanced homodyne detector, and find that remaining instability all occurs in a frequency region $\ll 1$ kHz, which does not affect the dynamics of our particles (except perhaps in point 5 below). We include a graph of our phase noise, which although not calibrated, clearly shows the low-frequency nature of the noise.

We have added “The relative phase between the two beams is monitored, but not stabilized, and only has significant noise below 100 Hz, far from any resonance of our trapped particles.” to the **Experimental system** section of the **Methods**.

Reviewer 1 states:

“4. The important experimental parameters, such as the power of the trapping beams, the optical aperture and the focal length of the high NA lenses, are missed, which are much useful for the readers to estimate the experimental results.”

Our answer:

Thanks for the referee’s comments. We have added more experimental details in the **Experimental system** section of the **Methods**: “As illustrated in Fig. 3a, a pair of 1550 nm Gaussian beams (~ 0.6 W) are focused by two NA = 0.6 lenses (clear aperture 3.60 mm, effective focal length 2.97 mm, Thorlabs 355660-C), such that their foci coincide, and a silicon nanorod is levitated in the antinodes of the resultant standing wave field.”

Reviewer 1 states:

“5. The signal shapes of the rotation in Fig.3.d is roughly rectangular, which is much different with the signal of a common optically-levitated rotor in vacuum (it is usually a Lorentzian function). Are there any other effects? it is better for the author to provide an explanation. In addition, from the usual point of view considering the focused beam, f_{β} is smaller than f_{α} , however, in Fig.3.e, f_{α} is larger than f_{β} , can the authors present an explanation?”

Our answer:

The Power Spectral Density (PSD) of rotation over a short time indeed shows a Lorentzian shape, but the data we take is averaged over 10 seconds, which is many millions of rotational periods. As discussed in other works [Kuhn *et al.* *Optica* **4**, 356 (2017), van der Laan *et al.*, *Phys. Rev. A* **102**, 013505 (2020)], since rotation is not a resonant phenomenon, the frequency depends sensitively on the local optical intensity and environmental factors like gas pressure (our work is at a somewhat higher

pressure than many in the literature). Drifts in any direction, including slow drifts that might be expected from the response to point 3, cause the rotation to decelerate and subsequently accelerate.

Below we include data showing the rotation PSD from the manuscript (red) taken over 10 seconds, with two superposed rotation PSDs taken over 0.2s, clearly illustrating that on short timescales the rotation indeed has a Lorentzian form.

We have added “The f_{rot} spectral feature is broad due to low-frequency drifts in the system \cite{kuhn2017} over the 10 s in which this data was acquired.” to the manuscript where Figure 3 is discussed.

Regarding the relative values of f_{α} , f_{β} , we note that in the literature other groups have either trapped oblate spheroids or nanodumbbells. We levitate thin rods in the Reyleigh-Gans regime, with length $L = 880$ nm, and diameter 210nm. In this case, the ratio of the frequencies is [Kuhn *et al. Optica* **4**, 356 (2017)]:

$$\frac{f_{\beta}}{f_{\alpha}} = \sqrt{\left(1 + \frac{X_m}{\Delta X} \frac{kL^2}{12}\right)}$$

where X_m is the maximal susceptibility, ΔX the susceptibility anisotropy, and k the wavevector of the optical field. Since all terms are positive, this ratio is always greater than 1 for such a geometry, and hence we always expect $f_{\beta} > f_{\alpha}$.

Reviewer 1 states:

“6. Some other important works of the longitudinal SAM are missed, for instance: Reimann, R. et al. GHz rotation of an optically trapped nanoparticle in vacuum. Phys. Rev. Lett. 121, 033602 (2018). Ahn, J. et al. Optically levitated nanodumbbell torsion balance and GHz nanomechanical rotor. Phys. Rev. Lett. 121, 033603 (2018).”

Our answer:

We thank the referee’s suggestion of the important references. In the previous version, we cited the first reference as [37] R. Reimann, et al., Phys. Rev. Lett. 121, 033602 (2018); for the second reference, we cited the latest reported results of rotation of levitated nanodumbbells from the same group [21] J. Ahn, et al., Nature Nanotechnology 15, 89 (2020). We added the second references in the manuscript as [24] J. Ahn, et al., Phys. Rev. Lett. 121, 033603 (2018).

2 REVIEWER 2

Reviewer 2 states:

“This is a well written paper describing new results on the experimental observation of the effects of transverse orbital angular momentum on a nano-rod optically trapped by two tightly-focused counter-propagating parallel but offset beams with identical linear polarization. The observed torque is orders of magnitude stronger than that observed in previous groups, with \sim MHz rotation speeds achieved at \sim mbar pressures, in part due to the geometric structure of the nanorod. The authors make a convincing case that what they observe is actually rotation from TAOM and not due to spin angular momentum e.g. due to birefringence in the material and imperfect laser polarization. The work is novel and will be a useful addition to the field, potentially enabling applications in torque sensing and rotational sensing, including possibly investigating effects such as Casimir effects and quantum friction, and I recommend publication.”

Our answer:

We appreciate the reviewer for her/his positive assessment and the recommendation of our work for publication.

Reviewer 2 states:

“I have a couple of minor comments on Fig. 3 manuscript.

1. The horizontal axis for Fig.3 (d,e) is labeled in units of Hz. I assume this is a typographical error and this should be kHz?”

“2. The lowest tick marker on the vertical axis of Fig. 3(i) is labelled as 10^1 MHz. I assume this should be 10^{-1} MHz.”

Our answer:

We thank the reviewer careful check and apologise for the typos. We have corrected the typographical errors in Figure 3, and double checked the revised version.

REVIEWERS' COMMENTS

Reviewer #1 (Remarks to the Author):

The authors of manuscript No.396805 have well replied to the questions. Especially, they present a concise explanation of the mechanism of the transverse rotation for a nanorod trapped by two counter-propagating laser beams and illustrate the equivalence of the so-called “out-of-center pushing force”, and the transverse OAM. They also explicate the difference between a nanosphere and a nanorod in an optical trap, the latter in the trap can be stably trapped without jumping between the adjacent antinodes. A supplemental low-frequency noise spectrum of the passive vibration isolation system demonstrates the stability of the trap. The explanation about the line shape of the rotation signal is credible. And the necessary experimental parameter as well as some references are added. At the same time, the other problems raised by the other referee are also revised. Based on the new version, I think this paper matches the high-quality requirement of Nature Communications, which provides a novel system for studying the transverse OAM, I recommend this paper can be published.

Reviewer #2 (Remarks to the Author):

The authors have successfully fixed the typographical errors related to Fig. 3 and my original assessment that this is a well written paper describing interesting new results is unchanged. The newly added explanatory text and supporting materials in response to the questions raised by the other referee has further improved the clarity and readability of the manuscript and I recommend publication.